# Dual-Channel Transverse Fields Radiofrequency Coils for 1.5 T Magnetic Resonance Imaging

**DOI:** 10.3390/s24072049

**Published:** 2024-03-23

**Authors:** Giulio Giovannetti, Denis Burov, Marcello Alecci, Rocco Rollo, Angelo Galante

**Affiliations:** 1Institute of Clinical Physiology, National Research Council (CNR-IFC), 56124 Pisa, Italy; giulio.giovannetti@cnr.it; 2Department of Physical and Chemical Sciences, University of L’Aquila, 67100 L’Aquila, Italy; denis.burov@graduate.univaq.it; 3Stelar s.r.l., 27035 Mede, Italy; 4Department of Life, Health & Environmental Sciences, University of L’Aquila, 67100 L’Aquila, Italy; marcello.alecci@univaq.it (M.A.); rocco.rollo@gssi.it (R.R.); 5Gran Sasso National Laboratory, Istituto Nazionale di Fisica Nucleare, 67100 L’Aquila, Italy; 6Superconducting and Other Innovative Materials and Devices Institute, National Research Council (CNR-SPIN), Department of Physical and Chemical Science, University of L’Aquila, 67100 L’Aquila, Italy

**Keywords:** MRI, dual-channel RF coils, transverse field, decoupling, mutual inductance

## Abstract

This theoretical study presents the design and analytical/numerical optimization of novel dual-channel transverse fields radiofrequency (RF) surface coils for 1.5 T Magnetic Resonance Imaging (MRI). The research explores a planar setup with two channels on a row with aligned spatial orientation of the RF coils, aiming to solve a common design drawback of single-channel transverse field RF coils: the reduced Field Of View (FOV) along the direction of the RF field. A significant challenge in this design is the efficient decoupling of two sets of transverse field RF coils to prevent mutual interference. Our modeling approach integrates thin wire theoretical modeling, magnetostatic computation for strip conductor coils, and their full-wave electromagnetic simulation. Key findings at 64 MHz demonstrate that strategic geometric placement among the two-channel RF coil and the introduction of geometrical asymmetry in the design of the individual RF coils does minimize the mutual inductance, paving the way for effective dual-channel MRI applications. This decoupling approach allows to enhance the FOV, providing a theoretical framework for the development of optimized dual-channel transverse field RF coil configurations. The current design was validated with full-wave numerical study at 64 MHz (^1^H, 1.5 T), has the potential to be extended at lower or higher frequencies, and the presence of lossy samples needs to be considered in the latter case.

## 1. Introduction

Magnetic Resonance Imaging (MRI) radiofrequency (RF) surface coils may be classified by the nature of their RF magnetic field within a given Region Of Interest (ROI): RF coils with a primarily axial field (i.e., orthogonal to the coil’s surface) and RF coils with a primarily transverse field (i.e., parallel to the coil’s surface) [1]. The main differences between them are the compromise between Signal-to-Noise Ratio (SNR), field homogeneity, and constraints to their use, depending on the sample shape and static magnetic field orientation. 

Transverse RF coils typically show high SNR in a small ROI, while axial RF coils boast higher homogeneity in a larger ROI [2]. There are exceptions, however, mainly in the realm of metasurfaces for MRI, which show higher SNR than traditional phased-arrays RF coils with the drawback of reduced Field of View (FOV) [3]. Transverse RF coils are adopted for niche applications, albeit being highly customizable in terms of sensitivity profiles and spatial selectivity [4]. 

Figure-of-eight (FO8) or “butterfly” type RF coils are composed, as the name implies, of a single conductor in the shape of a number eight (Lemniscate of Bernoulli) with two straight intersecting elements [4,5]. Apart from the high SNR in a restricted ROI, these RF coils can lie below, above, or on the sample’s side for both solenoidal and transverse B0 field MR systems, configurations often not available for standard loop coils. This geometry generates a transverse magnetic RF field in the region above (and below) these central current paths, where the RF coil’s FOV is located. The remaining electrical paths, needed to connect the straight elements along them and to the RF port, are far away from the FOV region, thus providing a negligible contribution to the RF field. In a rectangular FO8 (rFO8) variant, which maintains the same topology and upon which we will focus, the crossing point is displaced laterally and the central elements are parallel and non-crossing. Within the limits imposed by the wavelength of the RF field and the total RF coil electrical length, the central current paths can be made as long as necessary together with the FOV along that direction. At the same time, the distance among these paths cannot be increased above a few centimeters to avoid (i) a highly inhomogeneous sensitivity profile along this direction close to the coil’s plane and (ii) a poor sensitivity at larger depths. This generally limits the sensitivity extension of such RF coils along the RF magnetic field (${\vec{B}}_{1}$) direction. A possible solution to this drawback is to add at least a second identical RF coil, shifted along the ${\vec{B}}_{1}$ direction, to increase the extent of the sensitivity region. 

In the past, transverse RF coils have been proposed as a single-channel detection element [6], or in combination with (i) an axial coil [7] or (ii) another coplanar transverse coil rotated by 90° degrees [8], to form a two-channel system where the two RF coils are geometrically decoupled and the respective RF fields are orthogonal to each other in the ROI, providing a circularly polarized ${\vec{B}}_{1}$ field that decreases transmit RF power demands and increases SNR [5]. 

In this work, we designed a novel two-channel setup made by having two transverse field RF coils with the same spatial orientation. We also considered geometrically asymmetric transverse RF coils, each composed by squared loops of different size. The numerical study at 64 MHz (^1^H, 1.5 T) allowed us to minimize the mutual inductance (i.e., optimize decoupling), paving the way for effective FOV enhancement and providing a theoretical framework for the development of optimized dual-channel transverse field RF coil configurations. In Section 2, we present a theoretical model of mutual coupling among transverse RF coils as well as introduce methods to calculate coil’s mutual inductance using a magnetostatic approach and the $S_{12}$ scattering parameter by full-wave simulation. In Section 3, we present the results obtained by the three methods. In Section 4, we discuss the results. Our conclusions are reported in Section 5. 

## 2. Materials and Methods

We will focus on the rectangular FO8 (rFO8) type of coil with parallel central current elements, which can be approximated as composed by two rectangular loops with counter-rotating currents (Figure 1a). When moving away from the central point between the coil’s center (point C in Figure 1a) along a direction orthogonal to the coil’s plane, the sensitivity profile generated by current paths reaches a maximum at a depth that is a function of their distance s. This design approximates a real rFO8 coil, which neglects the port and the connection element between the two loops. 

The first two approaches will be based on the magnetostatic limit and focused on the calculation of the geometrical mutual inductance among two rFO8 coils (Figure 1b). This is appropriate when the RF wavelength is much higher than the electric length of the coil [9], and such a limitation should be kept in mind when realizing coils for specific applications. The first approach will use the thin-wire approximation. The second approach assumes strip conductors with negligible thickness. The third one is based on full wave simulation of the rFO8 coil, realized with strip conductors and complete with all electrical connections. The latter will consider all the physical effects and provide an evaluation of the scattering parameters including the effect of load.

In the following text, we will always consider pairs of rFO8 coils with the same spacing s and use the central current elements to define the relative shift d (Figure 1b). This allows an easy generalization to the case of rFO8 constituted by two loops of different size along the horizontal direction. 

The results for squared loop coils will give qualitative insights valid also for the circular loop transverse coils, thus providing a general framework for understanding the coupling among two transverse coils in general.

### 2.1. Analytical Calculation of Mutual Inductance

From the classical paper [10], it is known that for two identical square RF coils it is possible to find the optimal shift between them (equal to 0.9 D, where D is the side of the square) to null their mutual inductance. Coil coupling is mediated both by magnetic and electric fields, but the former is generally the dominant term for MRI applications at low fields (≤1.5 T), and we will focus on it in this study [11]. Within this approximation, zero mutual inductance between two rFO8 coils allows for independent channels operation, crucial for parallel MRI techniques [12], and this condition avoids frequency splitting of the resonant coils.

To calculate the mutual inductance of two rFO8 coils, we start from the analytical expression for the mutual inductance [13] of two coplanar rectangular current loops of equal height positioned on a plane, as shown in Figure 2:(1)$$M=\mu_{0}/4\pi \cdot [K(0,m,l_{1}+s,l_{1}+s+l_{2},m)-K(0,m,s,s+l_{2},m)\;-K(l_{1}+s,l_{1}+s+l_{2},-m,0,l_{1})+K(l_{1}+s,l_{1}+s+l_{2},0,m,l_{1})],$$
where μ_0_ = $4\pi \cdot 10^{-7}$ Henry per meter (H/m) is the free space permeability,
$$K(Z_{1},Z_{2},r_{1},r_{2},L)=\sum_{i=1}^{2}\sum_{j=1}^{2}(-1{)}^{i+j}[f(Z_{i},r_{j},0)-f(Z_{i},r_{j},L)]$$
and
$$f(Z,r,L)=\sqrt{(Z-L{)}^{2}+r^{2}}-(Z-L)\cdot ln[(Z-L)+\sqrt{(Z-L{)}^{2}+r^{2}}].$$

In the above expressions, the dimensions and relative positions of the RF coils are as from Figure 2 with $m,l_{1},l_{2}>0$; s is positive (non-overlapping) or negative (overlapping); and infinitely thin wires are assumed.

From Faraday’s law of induction, it is evident that the mutual inductance M is positive and has a maximum when the two squares overlap ($s<-l_{2}$), becomes zero for some negative s value, has a minimum (with negative value) for s=0, and then approaches zero asymptotically from below as s goes to infinity.

We can approximate an rFO8 coil as two squared loops, one with current flowing clockwise and one counterclockwise (see Figure 1a), using the superposition principle for the calculation of mutual inductance among a pair of rFO8 coils. We chose odd numbers (1, 3) to index the loops belonging to the leftmost rFO8 coil and even numbers (2, 4) to index the loops of the rightmost one (Figure 1b). Expression (1) can be used to compute the mutual inductance $M_{ij}$(i ≠ j) among loop pairs belonging to different rFO8 coils, and the mutual inductance among the two rFO8 coils can be written as follows:(2)$$M=M_{12}-M_{14}-M_{32}+M_{34}.$$
where the sign is positive for pairs of loops with current circulating in the same direction and negative for pairs of loops with current circulating in opposite directions. For identical rFO8 coils in the aforementioned configuration, the following statements are true: (i) $M_{12}=M_{34}$; (ii) $M_{12},M_{34},M_{32}$ can be positive or negative, depending on the loops’ size and relative position; (iii) $M_{14}$ is always negative because loops 1 and 4 are always not overlapping for positive shifts.

Numerical calculations for the computation of Expressions (1) and (2) were carried out using custom code developed in MATLAB (The Mathwork Inc., Natick, MA, USA), provided as open-source software (see the Appendix A).

### 2.2. Magnetostatic Simulation of the RF Coils

The analytical approach introduced in the previous section has two limitations: (i) the thin wire approximation and (ii) the two coils belonging to the same plane, a non-realistic condition for practical MRI applications. Indeed, surface RF coils are commonly made with strip conductors printed on a dielectric substrate [14], which forces the second RF coil on a different plane. To consider the effect of conductors with non-zero dimensions, we performed mutual inductance calculations for strip conductors of negligible thickness. The simulations are implemented in IDL 6.0 (Interactive Data Language, Visual Information Solutions, Boulder, CO, USA).

The mutual inductance between two conductors carrying uniform current densities $J_{1}$ and $J_{2}$ in the volumes V_1_ and V_2_ can be estimated with the following expression [15]:(3)$$M=\frac{\mu_{0}}{4\pi I_{1}I_{2}}{∭}_{V_{1}}{∭}_{V_{2}}\frac{J_{1}(r)\cdot J_{2}(r\prime )}{R}dvdv\prime$$
where $I_{1}$ and $I_{2}$ represent the total currents in volumes V_1_ and V_2_, respectively, and R = |r-r′| (Figure 3).

The mutual inductance between two rFO8 coils, each represented with a couple of identical rectangular loops laying on parallel planes with the distance between them equal to h (Figure 4), can be calculated with Equation (3) by considering the direction of the currents flowing in the conductors.

### 2.3. FDTD Simulations

Full-wave simulations were performed with the Finite-Difference Time-Domain (FDTD) method using the commercially available software XFdtd 7.8 (Remcom, State College, PA, USA), which allows the simulation of RF coils with arbitrary geometries. We included a phantom simulating a biological tissue, allowing to estimate the effect of load on the decoupling values between the two channels [16]. In this work, all RF coil elements were designed using the geometry workspace of the XFdtd tool. The simulated rFO8 RF coils were constituted by a Perfect Electric Conductor (PEC) with 4 mm width strips (Figure 5). The simulations were performed at 64 MHz (^1^H, 1.5 T).

In the two-channel RF coil assembly, each coil has a 50 Ω RF port inserted through a 4 mm opening into the rightmost coil. The response to a 64 MHz sinusoidal waveform of amplitude 1A on the first port was used to determine the S_12_ scattering parameter in the absence of any tuning device. The non-resonant nature of the simulated RF coils, as well as the use of PEC instead of copper, prevents an accurate calculation of electric coupling but allows us to compute the magnetic coupling, which is the dominant term at the selected working frequency [11]. The relative distance d between the two rFO8 RF coils was adjusted in 1 mm steps, while the distance h between the two planes was fixed at 2 mm and the linear elements separation s was 1 cm.

The simulations were performed both in unloaded and loaded conditions, the latter with a parallelepiped homogeneous phantom (27.5 × 12.5 × 5 cm) whose dielectric properties met the ASTM (American Society for Testing and Material) criteria for MR phantom at 1.5 T (electrical conductivity σ = 0.6 S/m, permittivity ε = 80 F/m) [17]. Successively, the magnetic field B_1_ distribution was estimated by feeding a 64 MHz (1A amplitude) sinusoidal input to both coils’ ports.

## 3. Results

### 3.1. Analytical Solutions for Mutual Inductances of Two-Channel Symmetric rFO8 RF Coils

First, let us consider two identical rFO8 coils with sizes $l_{1}=l_{2}=4.5$ cm, m=10 cm, and s=1 cm. Since $l_{1}=l_{2},$ the two constitutive loops of each coil have equal size and we will refer to them as symmetric rFO8. The mutual inductances among the loops, as well as the total mutual inductance among the rFO8 coils, are reported in Figure 6. We observe that M=0 can be obtained when d=3.14 cm. The result demonstrates that decoupling can be obtained for two identical and symmetric rFO8 coils, but its practical applicability is questionable. The reason is that when d>2s, the central current elements of the two rFO8 coils are quite distant and the RF field sensitivity profile along the *x*-axis develops a void, making the whole system of little use for MRI applications. An example is reported in Figure 6b, showing an acceptable magnetic field B1 distribution in the central ROI (z = 10 cm) along the transverse x-direction for d = 2s = 2 cm, while a broad void is present at the center for d = 6s = 6 cm.

### 3.2. Analytical Solutions for Mutual Inductances of Two-Channel Asymmetric rFO8 RF Coils

After considering a symmetrical configuration (same loops size within each rFO8 coil), we introduced a scaling factor λ, which is a multiplier of the width l used to change the size of one constitutive loop in the rFO8 coil with respect to the other (Figure 7). Since $l_{1}\neq l_{2}$, we will refer to this coil as asymmetric rFO8. We use the extra degree of freedom introduced by the λ parameter to reduce the mutual coupling in a two-channel setup with identical but asymmetric rFO8. We modify the size of loops 2 and 3 (see Figure 7), with the aim to reduce the larger and positive contributions $M_{12}=M_{34}$, thus decreasing the mutual inductance and achieving M=0 for a smaller shift d along the x-axis compared to the λ=1 case. This novel configuration can be obtained with two identical rFO8 RF coils by just flipping the second upside-down with respect to the first; for this reason, it does not add complexity to the design and realisation (same tuning/matching circuit for both coils). The result is shown in Figure 8a, where we considered the same geometrical parameters of Figure 6 and set λ=0.45. We observe that in this case, the M=0 condition is obtained for d=2 cm, with a 36% reduction with respect to the λ=1 case.

We next considered the d=2s case. We consider this shift value of practical interest because it allows all the four straight central elements to be separated from each other by a constant distance s, thus roughly doubling the FOV along x compared to the single-channel rFO8 case. We wanted to know if the optimal M=0 condition can be effectively obtained or, in case of M0, the residual coupling can be reduced enough to make such a configuration of practical use. Using the parameters l=m=4.5 cm, s=1 cm, and sweeping through the values of the scaling coefficient λ ranging from 0 to 2, we obtain the total mutual inductance M values reported in Figure 8b. It is clear that, with λ≠1, mutual inductance can be reduced, although not nulled. Thus, even for the d=2s condition, the use of asymmetric rFO8 has the potential to reduce coupling.

### 3.3. Magnetostatic Modeling of the Two-Channel rFO8 RF Coils

To validate the analytical results, we used the magnetostatic approach to compute the mutual inductance between two identical and symmetrical rFO8 coils (l=4.5 cm, s=1 cm, λ=1,andh=2 mm) with w=4 mm wide strips as a function of the shift of their central current elements d. Results are reported in Figure 9, and we can see that zero mutual inductance is found for d=3.25 cm, in good agreement with the analytical result (d = 3.14) obtained from the thin wire approximation (Figure 6).

Experimentally, the coupling between the two channels of the RF setup is evaluated by means of the $S_{12}$ scattering parameter, which is not affected by the M value alone but rather by the coils’ coupling coefficient $k=M/\sqrt{L_{1}L_{2}}=M/L$ [18], where $L_{1}=L_{2}=L$ are the self-inductances of the two identical rFO8 RF coils: $S_{12}=S_{12}(k)$. To check how the asymmetry parameter λ modifies the coupling coefficient, we calculated k for a range of λ values. The self-inductances were obtained via Equation (3) by substitution $j_{1}(r)=j_{2}(r\prime )=j(r),v_{1}=v_{2}=v$. The results (see Table 1) show a reduction of the coupling coefficient when a geometric asymmetry is introduced (i.e., λ≠1), thus providing support to the analytical results.

### 3.4. The Two-Channel rFO8 RF Coils from FDTD Simulation

Using the same geometry as in Section 3.3, we computed the $S_{12}$ parameter as a function of d for f=64 MHz (Figure 10). For the unloaded rFO8 coils, coupling is minimized ($S_{12}=$−55 dB) for d=3.0 cm, close to the results obtained with both the magnetostatic (d=3.25 cm) and analytical (d=3.14 cm) approaches. As shown in Figure 10, at the frequency considered in the simulations, the load has a small impact on the $S_{12}$ values, with a tiny displacement of the optimal decoupling condition towards larger shifts (unloaded: $S_{12}=$−54.9 dB at d = 3.0 cm; loaded: $S_{12}=$−51.2 dB at d = 3.1 cm).

We also note the presence of a second decoupling minima ($S_{12}=$ −75 dB) for d≃9 cm, which is consistent with the second zero of M from the analytical model (Figure 6). For this large shift value, $M_{12}=M_{34}$ and $M_{14}$ are small, and the M=0 condition is realized because loops 2 and 3 are decoupled due to their partial overlap (i.e., $M_{23}≃0$). From Figure 10, we also note that for d=2s=2 cm, $S_{12}≃-20$ dB, a useful value for practical operation in MRI.

Table 2 reports the FDTD results for the same coils with d=2s=2 cm and different λ values.

### 3.5. RF Field Mapping of Two-Channel Symmetrical and Asymmetrical rFO8 RF Coils

Figure 11 shows the modulus of the $B_{1}$ field maps obtained from the FDTD simulations in the x–y plane for z = 10 mm above the upper coils plane. We observe that the central region, corresponding to the four straight parallel current elements with spacing s, becomes less homogeneous for increasing asymmetry index values (smaller λ values) due to the closer proximity of the return paths.

To visualize the RF magnetic field distribution, we calculated the profile of the real part of the x component of the B1 field along a line parallel to the *x* axis that halves the coils (y = 0) for z = 10 mm. The results for different λ values are reported in Figure 12, where it is evident that the central lobe amplitude is reduced when large coils’ asymmetry is present (λ = 0.2, 0.4).

## 4. Discussion

From the previous sections, we provide evidence that two planar identical rFO8 RF coils can be fully decoupled when the central linear elements of the two coils are properly shifted with respect to each other, a condition that depends on the details of the coils’ geometry. This resembles a similar property of axial-field RF coils, widely used to decouple the nearest neighboring elements in a classical RF phased-array configuration.

Results from the analytical model (Figure 6 and Figure 8) are in reasonable quantitative agreement with both the magnetostatic calculations performed assuming strip-like conductors (Figure 9) and the full-wave FDTD simulations, obtained with or without a loading sample having electrical parameters equivalent to muscle tissue at 64 MHz (Figure 10).

Our first main result is the evidence of two different mutual shifts that decouple a two-channel setup made by two identical and symmetric rFO8 coils. The analytical model allows us to obtain some insights for both situations. The first one is that the $M_{14}$ contribution is always negligible due to the distance between the respective loops. For the smaller shift, we have a cancellation among the $M_{12}=M_{34}$ contributions and the $M_{23}$ one, with opposite signs. For the larger shift, $M_{12}=M_{34}≃0$ because the corresponding loops are well separated, while $M_{23}≃0$ because of critical overlap among loops 2 and 3 (see Figure 1b).

We conclude that the analytical model is a useful tool for the initial physical understanding of the RF coil design and the optimization steps required for a full design of the dual-channel rFO8 configuration at 64 MHz. For these reasons, we freely provide the custom-made MATLAB code for the calculation of the mutual inductance: it can be used and easily adapted by other groups to optimize the RF coils’ geometry according to their specific applications.

The zero mutual inductance condition guarantees zero (magnetic) coupling among the two RF channels. However, the geometrical constraints it introduces (i.e., the shift value that realizes such condition, $d_{M=0}$), can be unfit to the transverse-axis FOV target applications. This could be the case when $d_{M=0}>2s$, a condition that is associated with a sensitivity void along the transverse-axis direction.

We explored, for the first time, a possible geometrical modification of the rFO8 RF coil used in the two-channel setup, with the aim to reduce the decoupling shift, thus avoiding the RF sensitivity void. This can be achieved by properly positioning the return linear current paths of each rFO8 RF coil (i.e., the ones that are away from the central area close to the coil’s FOV). Among all possibilities, we considered the simplest one, introducing the asymmetry parameter λ that changes the size, along the transverse-direction, of one of the two loops that constitute each rFO8 RF coil (Figure 7). To stick closer to a practical realization of the proposed setup for 1.5 T MRI, we decided to consider two identical rFO8 RF coils with a clear advantage: the same design and tuning/matching capacitors can be used. The analytical model showed that the condition λ≠1 can indeed significantly reduce the shift value $d_{M=0}$ (Figure 8a); further, if needed, it can be used to reduce the rFO8 coils’ shift.

We finally focused on the two-channel setup made by two rFO8 RF coils with d=2s, which, compared to the single rFO8 case, should guarantee an almost doubled FOV with no voids in the RF field sensitivity profile. In this case, we changed strategy, shifting away from the M(d=2s)=0 condition, which previous results showed not to be attainable for the considered geometries. Here, we aimed to modify the coils’ geometry to minimize coupling. In practice, this means that $S_{12}$ should be less than −15 dB: $S_{12}\leq -15$ dB is commonly considered an acceptable inter-channel isolation level by the MRI practitioners, keeping in mind that additional preamplifier decoupling techniques [10,19] are typically applied.

The analytical model confirms that, by properly adjusting λ, the total mutual inductance M may be reduced (Figure 8b). To translate this result in terms of $S_{12}$, it is important to note that the M value was reduced more than the self inductance L, i.e., the magnetic coupling coefficient k was effectively reduced. This was confirmed by the magnetostatic simulations (see Table 1) and the final proof comes from the full-wave FDTD simulations reported in Table 2, showing that the S_12_ parameter improves from about −19 dB to −28 dB as λ varies from 1.0 to 0.2.

From the full-wave FDTD results obtained for symmetric rFO8 RF coils (Figure 10) we notice that S_12_ < −15 dB is satisfied for all shift values d≥2 cm. This suggests that, within the considered geometry, a two-channel setup configuration seems feasible from the point of view of coils’ decoupling even for λ=1. However, is worth noting that transverse field RF coils present return current paths with opposite current, as compared to the central conductive elements. In a multiple elements array configuration, the magnetic field contribution from the return paths will inevitably spoil the RF $B_{1}$ homogeneity in the central FOV, unless very large l values are considered for the building block RF loops. We conclude that a careful quantitative analysis of multiple-channel RF array configurations’ RF magnetic field profiles is necessary before stating their practical feasibility.

The use of multiple independent MRI receivers is adopted to enlarge the FOV and/or to allow parallel MRI for signal acquisition acceleration [5]. In both cases, the RF magnetic field sensitivity profiles of each RF coil element in the array should ideally have some degree of overlap to guarantee a relatively uniform coverage of the extended FOV. The results presented in Figure 11 demonstrate that this can be reasonably achieved with two rFO8 RF coils. A quantitative analysis of the RF $B_{1}$ homogeneity for the proposed setups is beyond the scope of this work since our focus was to prove the feasibility of RF coils’ decoupling.

Symmetric FO8 RF coils can operate in conjunction with (axial) loop RF coils. Such configurations can be designed to exploit geometrical decoupling (when the two coils have coincident centers) and quadrature operation. This is possible because both FO8 and loop coils have a symmetry plane parallel to the y axis but with opposite parity of the magnetic field along the *x* axis: decoupling is realized when such planes coincide. When a two-channel transverse field RF coil (both symmetric and asymmetric) is considered, a symmetry plane still exists for the total magnetic field, but it does not coincide with the symmetry plane of each rFO8 coil. We conclude that two rFO8 RF coils on a row can be decoupled but this prevents the geometrical decoupling with overlapping (axial) loop coils.

## 5. Conclusions

In this work, we proposed a two-channel transverse field RF coil setup that provides an extended FOV compared to the single-channel RF coil. The individual RF coils were designed to minimize the mutual coupling between the channels, also considering a specific geometric arrangement that should guarantee a good magnetic field homogeneity.

It represents a novelty since the use of transverse field RF coils in MRI was traditionally limited to a single element. When dual-channel setups were proposed, they consisted of either (i) an FO8 coil and a loop (axial) coil or (ii) two orthogonal FO8 coils. In both cases, the two coils were geometrically decoupled and, with orthogonal $B_{1}$ fields, used purposely to generate a circularly polarized field.

Our results are based on rectangular FO8 coils, but we expect the qualitative picture to remain valid if we consider different kinds of FO8 coils: (i) coils with semicircular shape return currents; (ii) FO8 with non-parallel but crossing central current elements and squared as well as semicircular return paths.

In our approach, a zero mutual inductance condition can be naturally realized, and, within some limitations, it can be tuned using the proposed asymmetry in the rFO8 RF coils’ design. This opens the possibility to consider array-like configurations of multiple rFO8 RF coils, where the nearest neighboring elements can be decoupled and the next-to-nearest ones could have small enough coupling to be operated directly or with the help of other decoupling methods (like preamplifier decoupling for receiver-only configurations). This could allow parallel MRI and signal acquisition acceleration in the future.

## Figures and Tables

**Figure 1 sensors-24-02049-f001:**
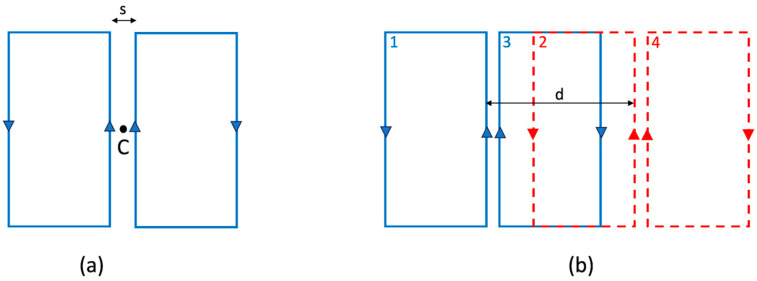
(**a**) Two rectangular RF coils with counter-rotating currents, used to schematize a rFO8 RF coil with symmetric rectangular return paths. Point C represents the geometrical center of the two central current paths separated by the distance s. (**b**) Two identical and symmetric rFO8 coils with the central current elements shifted by the distance d.

**Figure 2 sensors-24-02049-f002:**
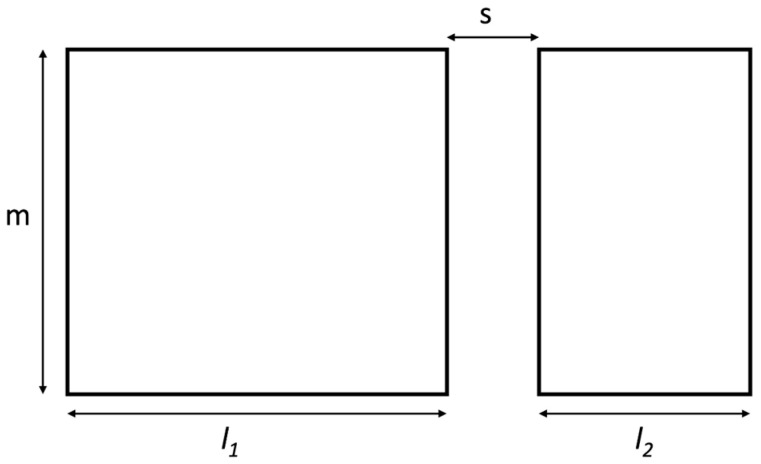
The geometric dimensions and relative position of two rectangular current loops.

**Figure 3 sensors-24-02049-f003:**
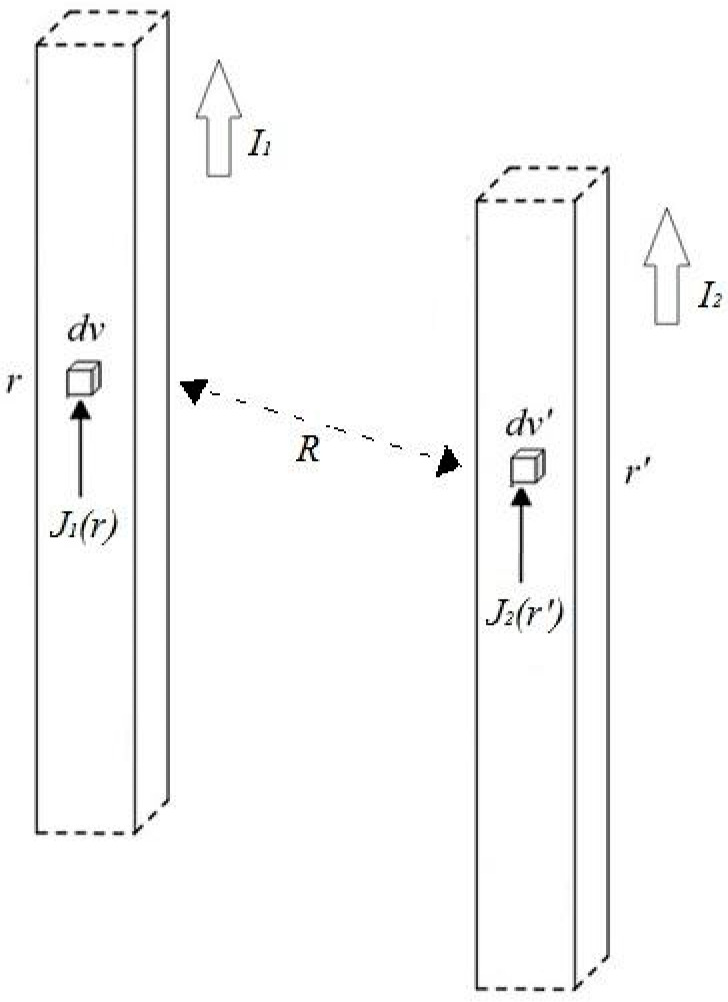
Conductor scheme adopted for magnetostatic mutual inductance calculations.

**Figure 4 sensors-24-02049-f004:**
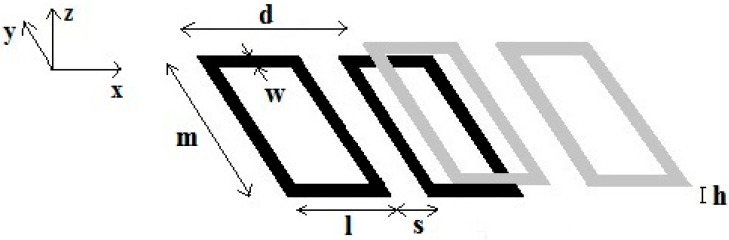
Sketch of two rFO8 coils realized with strip conductors. Dimensions m, l, s, d are as in Figure 1 and Figure 2; w is the strip conductor width; and h is the *z*-axis distance between rFO8 planes.

**Figure 5 sensors-24-02049-f005:**
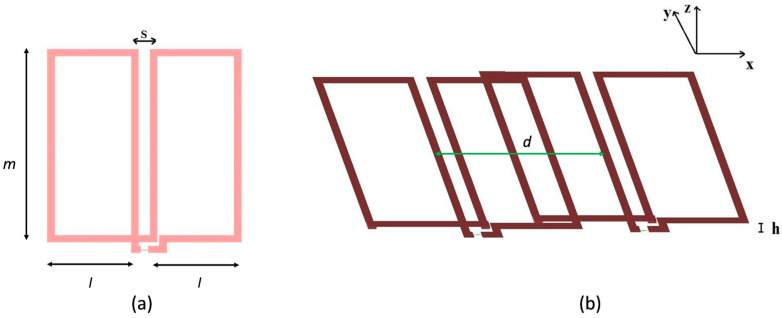
The model of (**a**) a single-channel planar rFO8 RF coil and (**b**) two-channel planar rFO8 RF coils with extended FOV along the *x*-axis.

**Figure 6 sensors-24-02049-f006:**
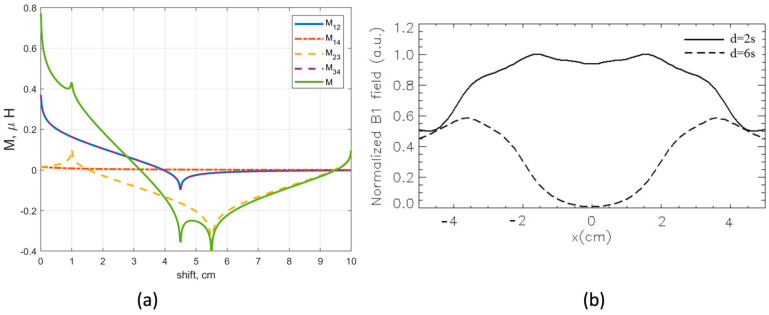
(**a**) Analytical calculation of mutual inductance between constitutive loops $M_{12}=M_{34}$, $M_{14}$, and $M_{32}$, and total mutual inductance M as a function of shift d among two identical and symmetric rFO8 coils with l=4.5 cm, m=10 cm, and s=1 cm. (**b**) The magnetostatic field profile at z = 10 mm along the *x* axis for the above configuration: d = 2s = 2 cm (continuous) and d = 6s = 6 cm (dashed).

**Figure 7 sensors-24-02049-f007:**
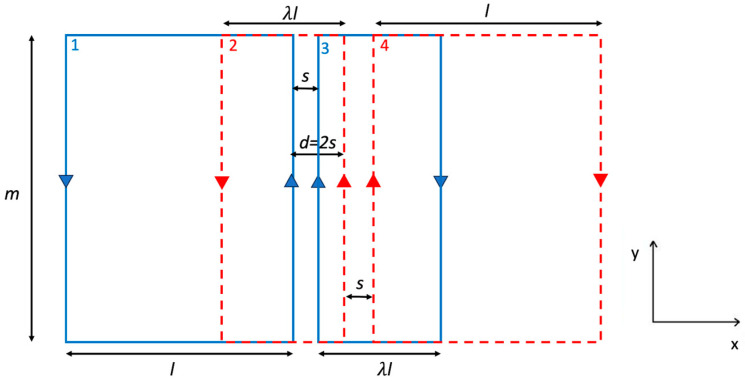
The model of a two-channel RF setup made by two identical and asymmetrical rFO8 coils with loop widths along the x-direction equal to l and λl, and a relative shift d = 2s that brings all the four central current paths to be equally spaced.

**Figure 8 sensors-24-02049-f008:**
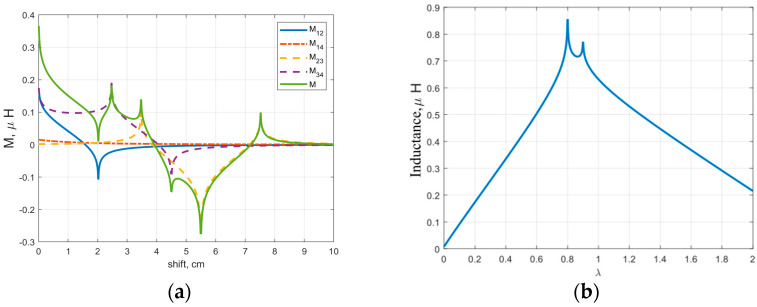
Analytical calculations: (**a**) total mutual inductance M and its components for two rFO8 coils with l=10 cm, m=10 cm, s=1 cm, and λ=0.45; (**b**) total mutual inductance M as a function of λ for l=4.5 cm, m=10 cm, and s=1 cm.

**Figure 9 sensors-24-02049-f009:**
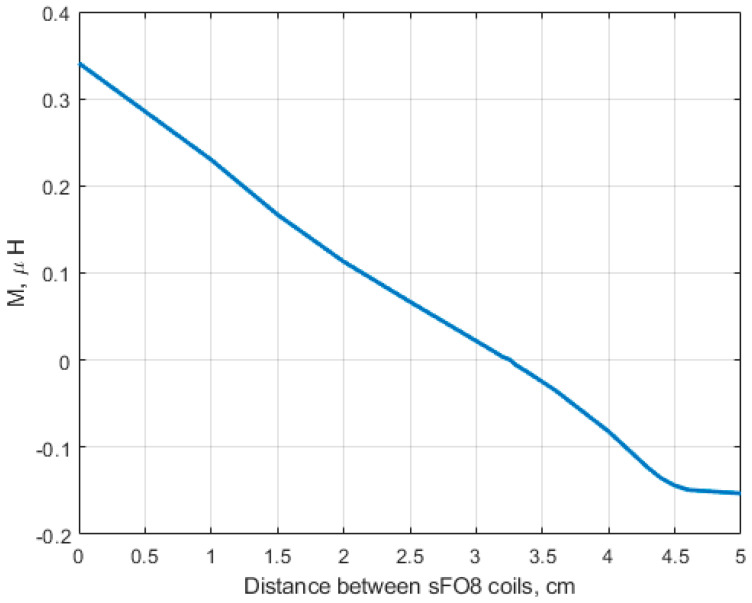
Magnetostatic computation of the mutual inductance of a two-channel RF setup made by two identical and symmetrical rFO8 coils (l = 4.5 cm, m=10 cm, s=1 cm, h=2 mm, and λ=1) as a function of the distance d.

**Figure 10 sensors-24-02049-f010:**
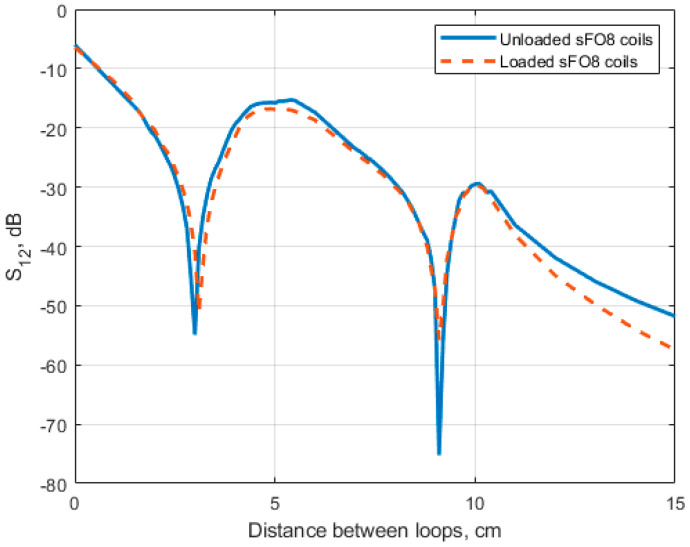
FDTD simulations: loaded and unloaded S_12_ versus the element distance between the two identical and symmetrical rFO8 RF coils (l= 4.5 cm, s=1 cm, h=2 mm, and λ=1) at 64 MHz.

**Figure 11 sensors-24-02049-f011:**
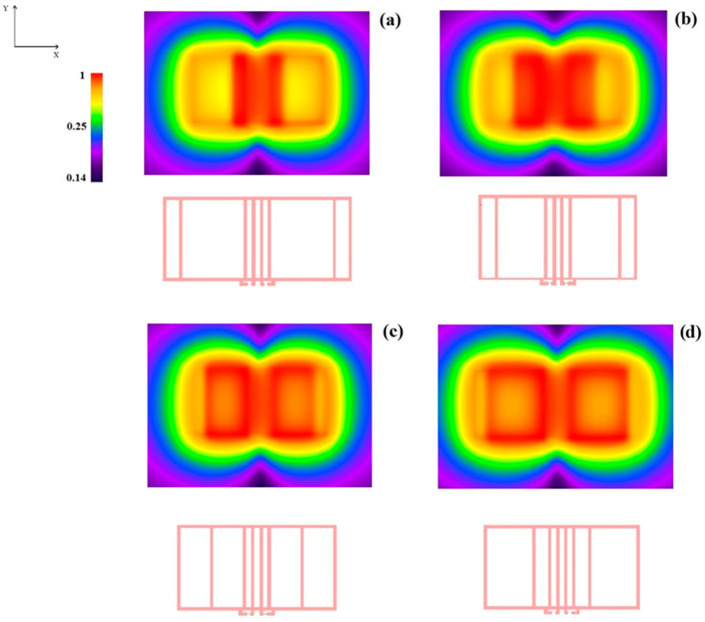
FDTD simulations: nodulus of the $B_{1}$ magnetic field profile (a.u., in dB) in the x–y plane at z = 10 mm for a two-channel RF setup made by two identical rFO8 RF coils (l=m=10 cm, d=2s=2 cm, and h= 2 mm): (**a**) λ=1.0, (**b**) λ=0.6, (**c**) λ=0.4, (**d**) λ=0.2.

**Figure 12 sensors-24-02049-f012:**
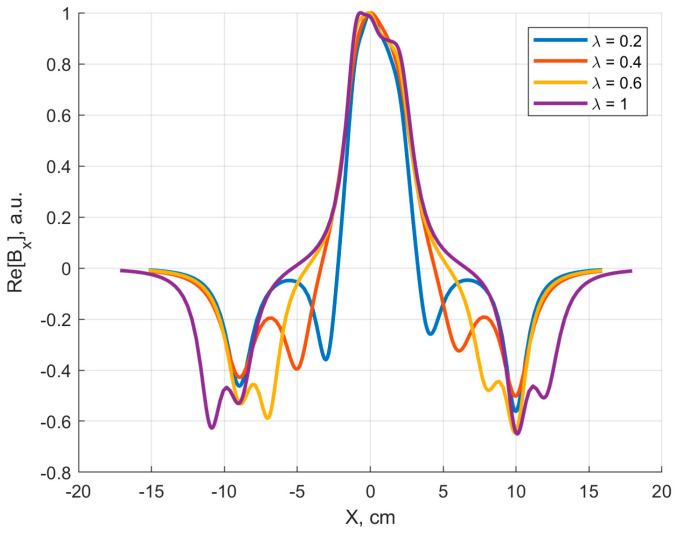
FDTD simulations: real part of the B_1,x_ magnetic field (normalized to 1 at maximum) for the four configurations of Figure 11 (z = 10 mm, y = 0). The plot is not symmetric due to the difference in z coordinates of the two coils’ planes.

**Table 1 sensors-24-02049-t001:** Magnetostatic calculation of mutual, self-inductance, and coupling coefficient of a two-channel RF setup made by two identical rFO8 RF coils as a function of the asymmetry index λ.

Asymmetry Index λ	M (nH)	L (nH)	k=M/L
1.4	289	726	0.39
1.0	314	677	0.46
0.6	240	598	0.4
0.4	154	552	0.29
0.2	77	491	0.16

**Table 2 sensors-24-02049-t002:** FDTD simulations: coupling coefficient of a two-channel RF setup made by two identical rFO8 RF coils, as in Figure 10 (d=2s=2 cm), as a function of the asymmetry index λ.

Asymmetry Index λ	Condition	S_12_, dB
1.0	Unloaded	−19.9
	Loaded	−19.4
0.6	Unloaded	−16.6
	Loaded	−18.0
0.4	Unloaded	−20.6
	Loaded	−22.1
0.2	Unloaded	−26.4
	Loaded	−28.7

## Data Availability

Dataset available on request from the authors.

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
