# Peer review of "Dual-Channel Transverse Fields Radiofrequency Coils for 1.5 T Magnetic Resonance Imaging"

_sensors, 2024, doi:10.3390/s24072049_

Round 1
Reviewer 1 Report
Comments and Suggestions for Authors
In this paper, the design and analytical/numerical optimization of novel dual-channel transverse fields radiofrequency (RF) surface coils for MRI application is reported. Normally, use of transverse field RF coils in MRI was traditionally limited to a single element, and its FOV is relatively small. The authors propose dual-channel transverse fields RF surface coils whose two channels are nearly in the same plane and decoupled with their geometrical asymmetry design, which extends the FOV and paves the way for its application in MRI. The method is of novelty and the demonstration is of sufficiency. I would like to recommend the publication of this work in sensors.
(1) Here is a suggestion that some statements which are not common sense or popular knowledge should be argued or cite literature. Two examples are listed as follows:
(i) In line 111-113, the authors state that “but the former is generally the dominant term for MRI applications at low fields (≤1.5 T), and we’ll focus on it in this study”. The statement that magnetic fields dominate in coils coupling is essential for this article, but no references are cited. Please supply more evidences or references to surpport this statement.
(ii) In line 207-209, the authors state that “The reason is that when 𝑑 > 2𝑠 the central current elements of the two rFO8 coils are quite distant and the RF field sensitivity profile along the x-axis develops a void, making the whole system of little use for MRI applications.” Evidences or references for this statement are required.
(2) In line 325-328, the authors state that “We observe that the central region, corresponding to the four straight parallel current elements with spacing s, becomes less homogeneous for increasing asymmetry index values 𝜆 due to the closer proximity of the return paths.” There is no obvious difference in the magnetic field non-uniformity in the central region of the four pictures in Figure 11. Maybe a curve that the modulus changes of the B1 field along the axis of the central region under four asymmetry index values 𝜆 will help the readers to understand.
Author Response
Please find the Reply in the attached PDF file

Reviewer 2 Report
Comments and Suggestions for Authors
Great work, and I only have some minor concerns.
1. Introduction to Figure-of-Eight (FO8) Coils:
The work primarily focuses on the figure-of-eight coil, deviating from the conventional circular or rectangular loop commonly used in RF coils. It would be beneficial for the authors to elaborate more in the introduction on why the figure-of-eight coil is significant in the context of RF coil design. Highlighting the advantages or unique features of FO8 coils that make them crucial for this study would provide readers with a clearer understanding of the research motivation.
2. Consideration of Coil Tuning/Matching Circuit in FDTD Simulation:
In the FDTD simulation to use S21 to evaluate coil coupling, it is crucial to ascertain the coil tuning/matching circuit was taken into account. Please provide details about how the coil was tuned and matched.
3. Coupling Between Loop Coil and FO8 Coils in Multi-Block Scenarios:
Since FO8 coils typically operate in conjunction with normal loop coils, it is pertinent to address the coupling when multiple blocks are used together. Specifically, the discussion section could explore how the coupling between the loop coil and adjacent fO8 coils is affected when two or more blocks are employed simultaneously, which is crucial for practical applications.
Author Response

(The authors gave the same response as above.)
